# Learning with Muscles: Benefits for Data-Efficiency and Robustness in Anthropomorphic Tasks

**Isabell Wochner**[*1]  **Pierre Schumacher**[*2,3]  **Georg Martius**[2]

**Dieter Büchler**[2]  **Syn Schmitt**[†1]  **Daniel F.B. Haeufle**[†1,3]

[1]Institute for Modelling and Simulation of Biomechanical Systems, University of Stuttgart, Germany
[2]Max Planck Institute for Intelligent Systems, Tübingen, Germany
[3]Hertie-Institute for Clinical Brain Research, University of Tübingen, Germany

**Abstract:** Humans are able to outperform robots in terms of robustness, versatility, and learning of new tasks in a wide variety of movements. We hypothesize that highly nonlinear muscle dynamics play a large role in providing inherent stability, which is favorable to learning. While recent advances have been made in applying modern learning techniques to muscle-actuated systems both in simulation as well as in robotics, so far, no detailed analysis has been performed to show the benefits of muscles when learning from scratch. Our study closes this gap and showcases the potential of muscle actuators for core robotics challenges in terms of data-efficiency, hyperparameter sensitivity, and robustness [2].

**Keywords:** reinforcement learning, model predictive control, actuator morphology

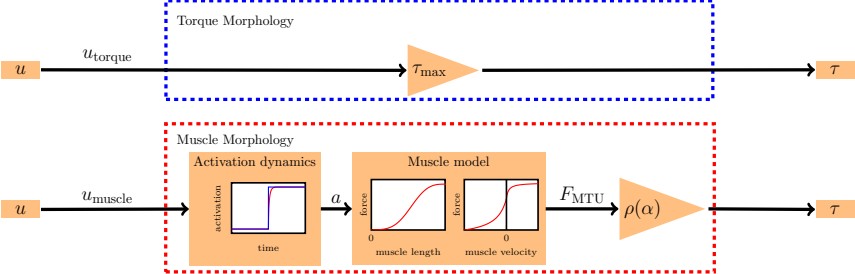

Figure 1: Key differences between torque actuator morphology and muscle actuator morphology.

## 1  Introduction

Recent developments in new learning methods allow the generation of complex anthropomorphic motions such as grasping, jumping or hopping in robotics. However, current systems still struggle with real-world scenarios beyond the narrow conditions of laboratory experiments. Humans, on the other hand, are capable of quickly adapting to uncertain, complex, and changing environments in a sheer endless variety of tasks. One key difference between biological and robotic systems lies in their actuator morphology: robotic drives are mostly designed to yield a linear relation between control signal and output torque. In contrast, muscles have complex nonlinear characteristics.

It has already been demonstrated, that muscular nonlinearities may provide a benefit for stability and robustness, especially under environmental uncertainties or perturbations [1, 2, 3]. A benefit over linear torque actuator morphology has been observed in computer simulations by exchanging the actuator morphology (similar to Fig. 1) in otherwise identical anthropomorphic tasks like reaching [4]

---

[*]Equal contribution. † Equal contribution.
[2]See https://sites.google.com/view/learning-with-muscles for code and videos.

6th Conference on Robot Learning (CoRL 2022), Auckland, New Zealand.

or locomotion [5, 6, 7, 8]. Similarly, reduced demand on the information processing capacity has been shown for muscles when compared to torque actuator morphology [9, 10, 11, 12, 13]. This opens the question whether muscular morphology could also be beneficial for robustness and data-efficiency in the process of *learning* movement control.

Recent advances in deep learning facilitated the generation of complex movements like point-reaching [14, 15, 16, 17] and locomotion [18, 19, 20, 21, 17] in simulations with muscular actuator morphology. In the real world, optimization and learning approaches can also find controllers for robotic systems with pneumatic muscles exhibiting somewhat muscle-like actuator morphology [22, 23]. These examples demonstrate that learning and optimization methods *can* control muscle-driven systems and may enable benefits such as safe learning and robustness [23]. However, investigating advantages of nonlinear muscular actuator morphology over linear torque actuator morphology requires a direct comparison of both, which is—to our knowledge—missing in the literature.

While Peng et al. [24] performed a comparative analysis of different actuator morphologies, their work was focused on replicating reference trajectories. In contrast, we learn behaviors without demonstrations, provide extensive hyperparameter ablations and not only employ RL, but also other optimization methods applied to complex 3D models.

The purpose of this study is to test if learning with muscular actuator morphology is more data-efficient and results in more robust performance as compared to torque actuator morphology when learning from scratch. We investigate this in a very broad approach: we employ different learning strategies on multiple anthropomorphic models for multiple variants of reaching and locomotion tasks solved in physics simulators of differing levels of detail. This provides new and comprehensive evidence of the beneficial contribution of muscular morphology to the learning of diverse movements.

## 2 Morphological difference between torque and muscle actuators

In contrast to idealized torque actuators, where torque is simply proportional to the control signal $u_{\text{torque}} \in [-1, 1]$,

$$\tau = \tau_{\text{max}} \, u_{\text{torque}} \tag{1}$$

muscular force output nonlinearly depends on the muscle control signal $u_{\text{muscle}}$, the muscle length $l_{\text{MTU}}$ and contraction velocity $\dot{l}_{\text{MTU}}$. These biologically observed dependencies can be predicted by so-called *Hill-type* muscle models [25]. In a nutshell, the model captures biochemical processes transforming muscle stimulation $u_{\text{muscle}} \in [0, 1]$ to the force-generating calcium ion activity $a$. This can be modeled by a first-order differential equation of the form [26]

$$\dot{a} = f_a(u_{\text{muscle}} - a) \tag{2}$$

which induces low-pass filter characteristics (Fig. 1). The model further captures the nonlinear *force-length* and *force-velocity relations* [25]. The *force-length relation* is characterized by a positive slope (increasing force with increasing muscle fiber length) in the typical operating range of biological muscle fibres (Fig. 1). The *force-velocity* relation is characterized by decreasing force for faster shortening velocities and increasing force if the muscle is externally stretched against its contraction direction (Fig. 1). A lever arm $\rho(\alpha)$ translates joint angle $\alpha$ into muscle-tendon-unit length $l_{\text{MTU}}$ and muscle force into joint torque

$$\tau = \sum_{i=1}^{N} \rho_i(\alpha) f_\tau \left( l_{\text{MTU},i}(\alpha), \dot{l}_{\text{MTU},i}(\dot{\alpha}), a_i \right). \tag{3}$$

for $N$ muscles which span a joint—typically at least two in an antagonistic arrangement.

In practice, we employ two different muscle models: A detailed one with more physiological details, contained in demoa [27], and a simpler model that efficiently adds muscular properties to existing MuJoCo [28] simulations without sacrificing computational speed. See Suppl. A for details.

## 3 Methods

### 3.1 Learning approaches for movement control

We test if muscle actuator morphology facilitates learning by applying state-of-the-art learning algorithms covering an extensive range of approaches currently used in robotics. The common thread of the selected algorithms lies in their independence of the actuator morphology: this allows us to easily exchange idealized torque actuator morphology with muscle actuator morphology. We choose optimal control, model-predictive control and reinforcement learning as learning approaches.

Table 1: **Overview of all models and tasks**

| Model | Task | Control | Environment |
|-------|------|---------|-------------|
| ArmMuJoCo | precise reaching | RL | MuJoCo |
| ArmMuJoCo | fast reaching | RL | MuJoCo |
| ArmDemoa | smooth point-reaching | opt. control, MPC | demoa |
| ArmDemoa | hitting ball with high-velocity | opt. control, MPC | demoa |
| Biped | hopping | RL | MuJoCo |
| FullBody | squatting | opt. control, MPC | demoa |
| FullBody | high-jumping | opt. control, MPC | demoa |

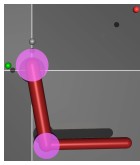 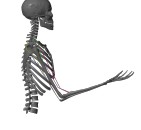 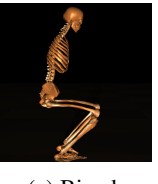 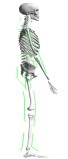

(a) ArmMuJoCo      (b) ArmDemoa      (c) Biped      (d) FullBody

Figure 2: Models used for the experiments.

**Optimal control (OC)** The control problem with horizon $N$ can be defined as:

$$\min_{\pi_k} J = \min_{\pi_k} \sum_{k=0}^{N} l(x(k), u(k), k), \qquad \text{subject to} \ \ x(k+1) = f(x(k), u(k), k),$$

$$u(k) = \pi_k(x(0), ..., x(k)). \qquad (4)$$

where $x(k) \in \mathbb{R}^{n_x}$ denotes the current state at time $k$, and $u(k) \in \mathbb{R}^{n_u}$ is the applied input at time $k$. Furthermore, $l$ specifies the cost function, and $f$ denotes the system dynamics. To optimize for the best control policy, we use the covariance matrix adaptation evolution strategy (CMA-ES) [29] in the optimal control case (open-loop strategy). CMA-ES is a derivative-free algorithm and widely used in machine learning. It combines different learning mechanisms for adapting the parameters of a multivariate normal distribution. Note, that we choose the same optimization parameters for both actuator morphologies to allow for a fair comparison even though the number of decision variables $n_u$ is always larger in the muscle-actuated case due to the antagonistic setup.

**Model predictive control (MPC)** In MPC, we solve the control problem in a receding-horizon fashion with varying prediction horizons and recursively apply only the first element of the predicted optimal control sequence $u(0)$ (closed-loop strategy). We employ a warm start procedure using the CMA-ES optimizer and afterwards start the MPC routine with the local optimizer BOBYQA [30]. The parameters of the optimizers are either varied (see Sec. 4) or given in Suppl. B.

**Reinforcement learning (RL)** RL allows learning of reusable feedback controllers. Instead of *minimizing* a cost function (see Eq. 4), conventionally the discounted expected reward is *maximized*:

$$\max_{\pi} J = \max_{\pi} \mathbb{E} \left[ \sum_{k=0}^{N-1} \gamma^{k-1} r(k) \right] \qquad (5)$$

where $r(k)$ is the reward at time $k$, $\pi$ is a control policy and $\gamma \in [0, 1]$ is a discount factor such that long-term rewards are weighted less strongly. RL consequently solves a similar problem to MPC, but the resulting controllers are able to act in closed-loop fashion without being given an explicit prediction model. For the point-reaching tasks, we additionally employ goals $g$ characterizing the desired hand position, which then constitutes an additional dependence of the reward function. The aim of the learning process is to learn a controller policy $\pi(u(k)|x(k))$ that takes as input the current sensor values, or state x(k), and outputs a control signal, or action, $u(k)$ such that a task is solved. In practice, we use the RL algorithm MPO [31], implemented in TonicRL [32].

## 3.2 Models

**Arm** The Arm model (Fig. 2 a, b) consists of two segments connected with hinge joints (2 joints total) moving against gravity. The ArmMuJoCo [28, 17] model contains four muscles, two for each

joint. In the muscle-actuated case in ArmDemoa [33, 34], six Hill-Type muscles generate forces: two muscles for the shoulder and two for the elbow joint, plus two biarticular muscles acting on both joints. All joints are individually controllable.

**Biped**  We converted the geometrical model of an OpenSim bipedal human without arms [19] for use in MuJoCo. The model, consisting of 7 controllable joints (lower back, hips, knees, ankles) moves in a 2D plane. Each joint is actuated by two antagonistic muscles or one torque actuator.

**FullBody**  The FullBody model [35, 36] consists of two legs and an upper body including arms based on a human skeletal geometry. It consists of 8 controllable joints (ankles, knees, hips, lumbar and cervical spine) in 3D, and 14 movable joints in total including the arms. Each controllable joint was either actuated by two antagonistic muscles (muscle-actuated case) or by one idealized torque actuator (torque-actuated case). For more details, we refer to Suppl. C.

All models and their respective physics differential equations were solved with variable time step (max. time step 0.001s) in demoa and fixed time step (0.005s) in MuJoCo.

### 3.3  Objectives and rewards

We choose anthropomorphic movement objectives which are highly relevant for robotic applications. We expect that muscular actuator morphology provides benefits for such tasks. All task formulations allow application in muscle and torque actuator morphologies with an identical reward or objective function. For a precise formulation of the used functions and conditions, see Suppl. D.

**Smooth point-reaching**  This task encourages *smooth* point-reaching. Therefore, the objective minimizes the L2-error between the desired and current joint angle while penalizing the angle velocity and jerk to ensure a smooth motion. The desired angle is $90°$ for both the shoulder and the elbow joint, as this requires a large motion.

**Precise point-reaching**  Similar to [13], we incentivize reaching a random hand position in a pre-determined rectangle, while minimizing the distance of the end effector to the goal. We specifically add a reward term that gives a much larger reward for precise motions that reach the center of the target area. The episode does not terminate until a time limit of 1000 steps elapses.

**Fast point-reaching**  The same objective as for precise point-reaching is used, but the episode terminates if the target is reached, incentivizing reaching speed over precision.

**High-velocity ball serve**  A ball is dropped in front of the Arm model and the controller learns to hit the ball to achieve maximum ball velocity.

**Squatting**  This squatting objective is taken from [35] and encourages desired hip, knee, and ankle angles for a squatting position.

**Maximum height jump**  The objective for the high-jumping task is taken from [37] and maximizes the position and velocity of the centre of mass of the human body model at the time of lift-off. The model is initialized to start from a squatting position.

**Hopping**  We developed an objective based on the z-axis velocity of the center of mass (COM) of the system that encourages periodic hopping with maximum height. The episode terminates if extreme joint angles are exceeded.

## 4  Results

In the following, we present three major results for the investigated approaches and environments: (**1**) Muscle-like actuators in general improve data-efficiency compared to torque-actuators. (**2**) The investigated learning and optimization algorithms exhibit greater robustness to hyperparameter variations when applied to muscle-driven systems. (**3**) The motions and controllers obtained from the muscular morphology are more robust against force perturbations that were not present during learning. We average results over 5 and 8 random seeds for OC/MPC and RL respectively.

### 4.1  Data efficiency: Learning with limited resources

Robotics applications in real-world scenarios often suffer from limited resources, which holds true for training and inference time. Therefore, we investigate the advantages of muscle-like actuator morphology in terms of overall learning efficiency and temporal control resolution.

**Advantages of muscular morphology**  Smooth and precise point-reaching generally require more data with torque-driven systems, as seen in Fig. 3. The performance of the muscle actuator, in contrast to torque morphology, varies very little for different settings of the temporal control resolution $c$. Precise reaching with RL also results in stable performance with fewer training iterations, and a very

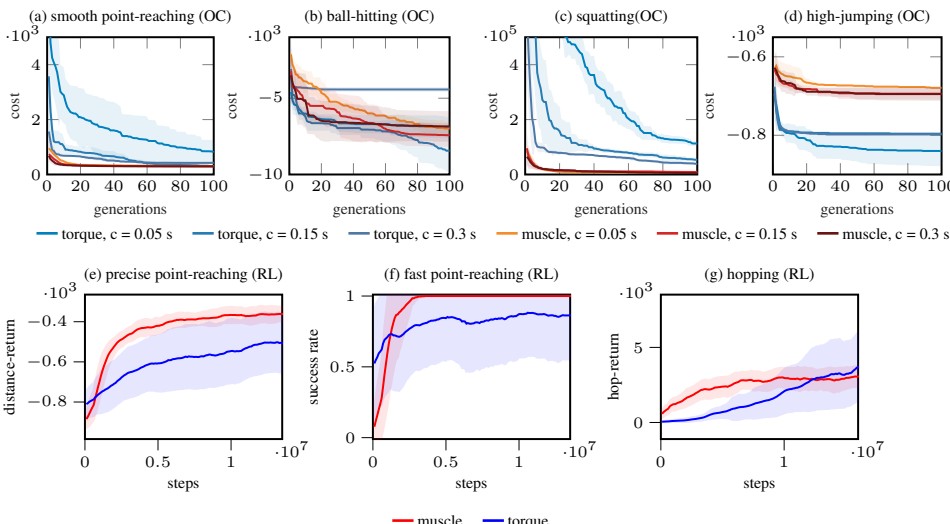

Figure 3: **Cost value or returns for different tasks.** Plotting the mean and standard deviation (shaded area) for 5 (OC/MPC) or 8 (RL) repeated runs for the two actuator morphologies (muscle in red, torque in blue). Additionally, the temporal control resolution $c$ was varied in the OC cases.

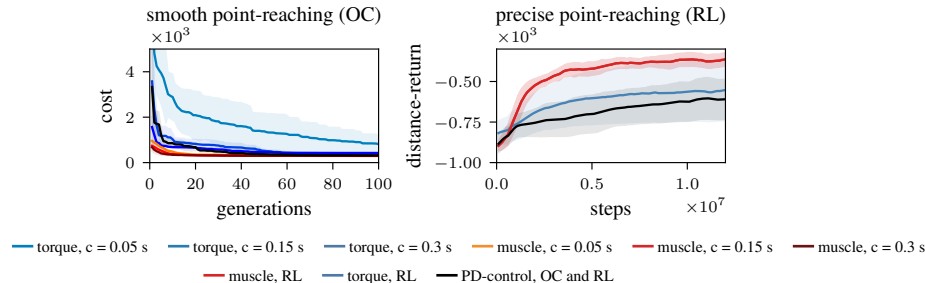

Figure 4: **Cost value or return in comparison with PD-control for torque.** Left: Muscles outperform all other considered morphologies with OC, while PD-control achieves lower cost than torque actuation with large control resolution $c = 0.3$. Right: PD-control does not yield an advantage over torque actuators with RL when applied to the precise point-reaching task.

small standard deviation across runs. Similar findings are seen for the squatting and hopping task, where muscle-actuators achieve better data-efficiency and smaller variation across runs and are able to find a good-enough optimum with fewer iterations.

**Advantages of torque morphology** In tasks requiring fast and strong motions, without emphasis on stabilization, we find torque actuators to hold certain advantages. In ball hitting and fast reaching, the torque cases show similar or smaller variance, even though both actuators perform well for singular runs. The high-jumping task, where only a strong, swift motion is required to launch the system upwards, is solved much faster in the torque case. We can also observe in the hopping task that, although only after a considerable number of training iterations and exhibiting a large variance, some torque-actuated runs achieve a larger overall return than the best muscle-actuated runs.

We additionally investigated a PD controller for the torque actuator morphology, see Fig. 4. While the PD controller slightly improves the data-efficiency for some cases, for both OC as well as for RL, the muscle actuator outperforms all baselines. See Sec. F.2 for more experiments.

### 4.2 Robustness to hyperparameter variations

Tuning a growing number of hyperparameters for learning models typically requires considerable time and computational resources. By analysing hyperparameter sensitivity, we test if tuning with torque or muscle actuator morphologies requires less resources.

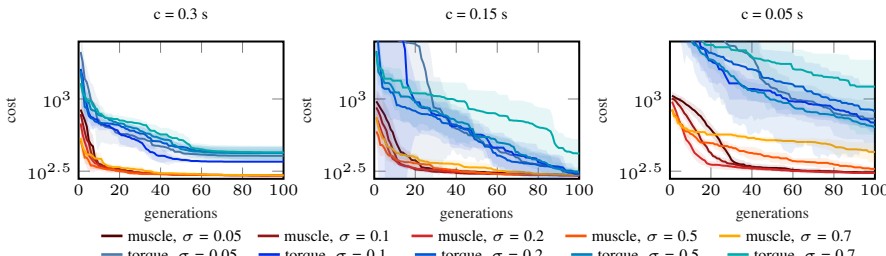

Figure 5: **Muscle morphology is more robust towards hyperparameter variation ($\sigma$) in point reaching.** The cost value of the best observation is shown. The mean and standard deviation (shaded area) are plotted for five repeated runs for the two actuator morphologies (muscle in red, torque in blue) with different control resolutions $c$.

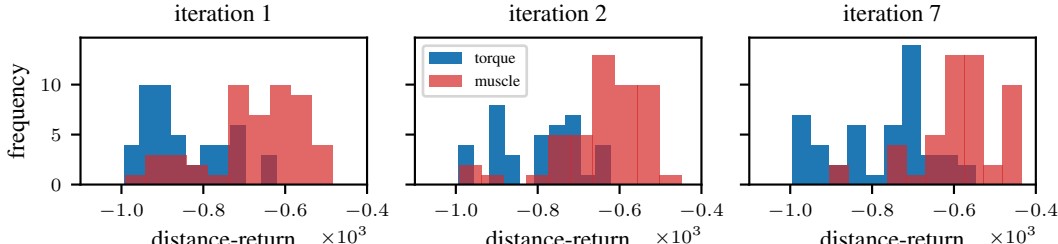

Figure 6: **Muscle actuators need less parameter tuning for good performance.** Hyperparameters are optimized for precise point-reaching following an iterative sampling scheme, each run trains for $2 \times 10^6$ iterations. Fifty sets of parameters are sampled randomly from pre-determined distributions, the final performance is evaluated and used to adapt the sampling distributions for the next iteration. We plot the return distributions over the sampled parameters at different iterations.

Figure 5 shows the cost curves for smooth point-reaching for the evolutionary optimization algorithm CMA-ES for different values of $\sigma$, which is the principal tuneable parameter for this algorithm. The performance curves vary much more for torque actuators for all considered cases. Furthermore, all muscle-actuated cases find a good-enough optimum with fewer iterations and a smaller variance, independent of the hyperparameter $\sigma$ and the control resolution $c$.

The same task was repeated using MPC while varying the main hyperparameter $t_{\text{pred}}$, which represents the prediction horizon in moving horizon strategies. The performance curves and the final cost vary much more for torque actuators (Fig. 7a, note, the cost is plotted logarithmically).

Finally, we performed an extensive hyperparameter optimization for precise point-reaching. For each iteration, 50 sets of parameters are randomly chosen and the final task performance is evaluated after $2 \times 10^6$ learning iterations. The sampling distributions for the parameters are then fit to the best performing runs and 50 additional sets are evaluated for the next iteration. We optimize the learning rates of MPO, as well as gradient-clipping thresholds, as these have a strong influence on learning speed and stability. Muscle actuators already outperform torque-actuators in the first iteration, with a greater number of well performing parameter sets (Fig. 6). Almost no low-performing runs remain for iteration 7, while a large torque-performance is only achieved by a small subset of parameter settings. See Suppl. E for more hyperparameter variations.

### 4.3   Robustness to perturbations

In this section, we probe the robustness of the obtained policies against unknown perturbations. In precise point-reaching, we evaluated the RL reaching policies for two modifications that were not present during training: First, the hand-weight of the model is increased by 1.5 kg (dynamic load), and secondly a free spherical weight is attached to the end effector with a cable (chaotic load). We can see in Fig. 8 that the muscle-based policy does not suffer significant changes in performance, except for a small circular motion (3 cm) around the goal position in the chaotic load case. In contrast, the torque actuator morphology leads to unstable reaching and strong oscillations. Both morphologies seem to handle the dynamic load well. See Suppl. E for more goals.

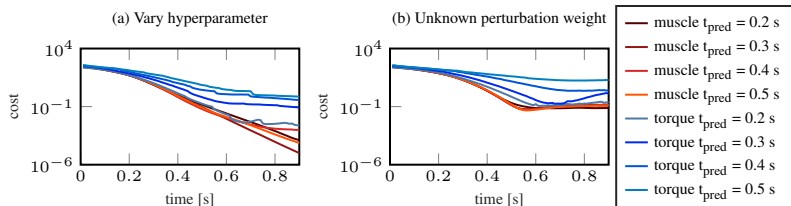

Figure 7: **Muscle morphology is more robust towards variation of hyperparameter $t_{pred}$ and unknown perturbations.** Plotting the development of cost over time for the two actuator morphologies (muscle in red, torque in blue) while varying the hyperparameter $t_{pred}$ denoting the length of the prediction horizon. Left: The unperturbed case. Right: The prediction model is not aware of the added weight to the lower arm.

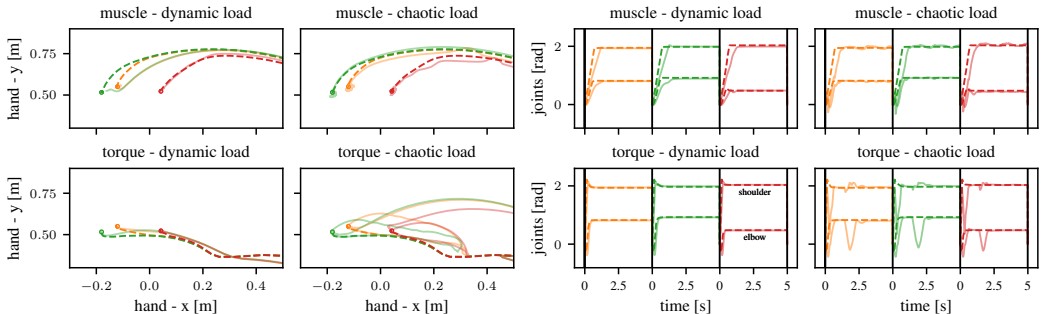

Figure 8: **Trajectories for dynamic (1.5 kg weight) and chaotic (attached ball) load.** Left: Three random goals are exemplarily shown, the respective goal position is marked as a circle. The unperturbed baseline for each goal is shown with a dashed line. Right: Joint trajectories for the same experiment, the unperturbed baselines are shown with dashed lines. Vertical bars mark episode resets.

For the MPC controller, we evaluated robustness by introducing environment changes that are unknown to the prediction model. One example is the lifting of an object with unknown weight, a typical robotics task. When adding 1 kg to the lower arm of the ArmDemoa model (Fig. 7), the performance in both actuator cases is worse than in the unperturbed case (left); the movement is also slower. However, the variance and absolute value of the final cost in the muscle-actuated case are still much lower compared to the torque-actuated case (plotted logarithmically). See Suppl. F.3 for more weight variations.

For periodic hopping with the Biped model, we evaluated trained RL policies with random forces that were drawn from a Gaussian distribution $F \sim \mathcal{N}(\cdot|0, \sigma_F)$ and applied to the hip, knee, and ankle joints with a probability of 0.05 at each time step. We see in Fig. 10 that the torque actuator morphology is stronger affected in relative performance than the muscle morphology. In the robustness investigation with MPC in the FullBody squatting task, a force is applied to the hip joint after the system has reached its desired position. Figure 9 (left) shows that the desired joint angles are much less affected by the perturbation when muscle actuators are controlled. Furthermore, the cost value associated with the movement recovers much slower for torque actuators (Fig. 9 right).

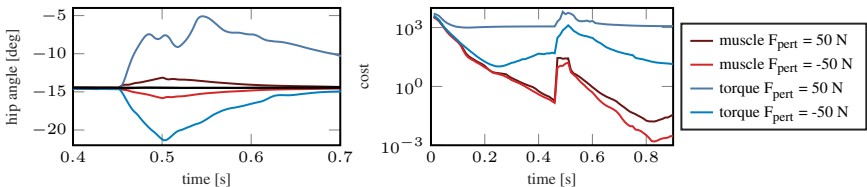

Figure 9: **Hip and cost trajectory for squatting with unknown perturbation forces.** The muscle morphology is shown in red, the torque morphology in blue while varying the perturbation force $F_{pert}$ [N] (applied between $t >= 0.45$ s and $t <= 0.5$ s).

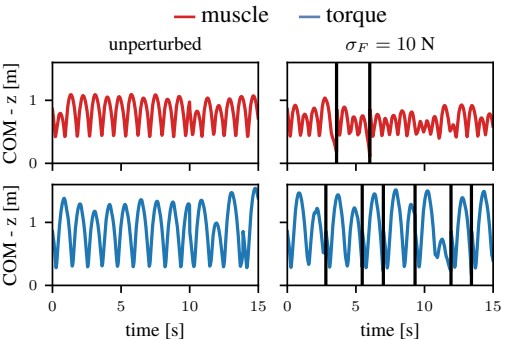
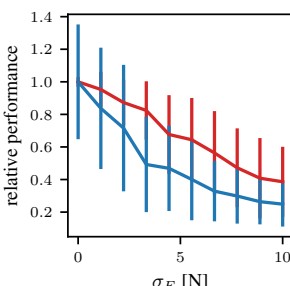

Figure 10: **Muscle actuators lead to more robust hopping.** At each time step, there is a 5 % chance of a random force being applied to the hip, knee, and ankle joints. Left: COM motion over time. Vertical bars mark episode resets due to falls of the Biped. Right: Relative performance for different standard deviations of the random force $\sigma_F$. Performance is scaled by the unperturbed mean return.

## 5 Discussion

We investigated if muscle-like actuators have beneficial effects for modern learning methods in terms of data-efficiency, hyperparameter sensitivity and robustness to perturbations. A multitude of variations across physics simulators, learning algorithms and task domains was considered in order to showcase the potential of the considered morphologies independently of the underlying implementation. We showed that muscles yield benefits in tasks requiring stable motion, even when compared to idealized torque actuators, which can be considered an upper performance bound. Indeed, the used torque actuators are able to instantaneously output any desired force at any point of the trajectory, while muscles only slowly change their output due to activation dynamics and can only produce kinematics-dependent force output. Despite these limitations, the considered learning algorithms learn more efficiently with muscle actuation in all tasks, except for extreme motions where objectives require a strong force application without stability considerations, such as ball-hitting and high-jumping. In bipedal hopping, it was found that muscles result in more efficient learning, even though some torque-runs achieve higher asymptotic performance. Finally, we observe muscle actuation to result in increased robustness to perturbations and hyperparameter variations, which can facilitate learning on real robotic systems that not only present sensor and motor noise, but also prohibit extensive parameter searches.

**Outlook for real-world robotics** We see two use-cases of our findings: (1) Muscular force-length-velocity and low-pass filter characteristics can be implemented as low-level actuator control for torque-controlled robotic systems (e.g., [38, 39, 40, 41]). This could allow us to exploit the improved data efficiency and robustness observed in our study for RL on a real robotic system. (2) Novel soft robotic actuators, such as artificial muscles [42, 43, 44, 45], promise to revolutionize specific application scenarios of robotics, e.g., wearable rehabilitation devices [46]. While soft actuated systems are hard to control from a classical control theory point of view, our results and other works [24] suggest that RL may even benefit from their properties. In our study, the simplified MuJoCo muscle model is applicable as a low-level controller in the sense of the first use case, while the results with the complex series-elastic muscle model in Demoa highlights the second use-case, making both cases strong arguments to consider RL and muscle properties a promising combination.

**Limitations** Although we have reported results for a wide variety of algorithms and tasks, we cannot give theoretical statements about the general applicability of our findings. Additionally, some of the tasks we employed were limited in complexity and might also be solvable with classical control algorithms. The MuJoCo muscle model, while computationally efficient, only captures rudimentary properties of biological systems. The demoa implementation, on the other hand, includes visco-elastic, passive tendon characteristics and muscle routing as joint angle-dependent lever arms to account for many physiological details—at substantial additional computational cost. Finally, learning with intermediate control signals given to impedance or position controllers, instead of direct torque commands, might also improve learning performance, while muscle-like properties could have been introduced by learning priors or additional cost terms.

**Acknowledgments**

We thank Daniel Höglinger for help during the development of the hopping reward function. Furthermore, we like to thank Marc Toussaint, Danny Driess and David Holzmüller for initial discussions regarding the topic of learning with muscles. This work was supported by the Deutsche Forschungsgemeinschaft (DFG, German Research Foundation) under Germanys Excellence Strategy - EXC 2075 - 390740016 (SimTech). We thank the International Max Planck Research School for Intelligent Systems (IMPRS-IS) for supporting all authors. Georg Martius is a member of the Machine Learning Cluster of Excellence, EXC number 2064/1 Project number 390727645. This work was supported by the Cyber Valley Research Fund (CyVy-RF-2020-11 to DH and GM). We acknowledge the support from the German Federal Ministry of Education and Research (BMBF) through the Tübingen AI Center (FKZ: 01IS18039B).

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
