# OpenReview forum: "Learning with Muscles: Benefits for Data-Efficiency and Robustness in Anthropomorphic Tasks"
_robot-learning.org/CoRL/2022/Conference — CoRL 2022 Poster_

### Official Review · Reviewer_KLzq · 2022-07-26

**Originality:** Good
**Technical Quality:** Fair
**Clarity Of Presentation:** Very Good
**Impact:** 2

**Recommendation:**

Weak Accept: I recommend accepting the paper, but will not argue for my recommendation if the majority of other reviewers have a different opinion.

**Summary:**

This paper argues that humans (and other animals) are able to perform a diverse range of motions *thanks to their muscle’s unique dynamical properties.* In particular, the authors hypothesize that "highly non-linear muscle dynamics” provide inherent stability that makes learning tasks easier.

To verify this, the authors test if calculating robot torques ($\tau$) from a control signal ($u \in [0,1]$) using a muscle activation model [1] (See *Eq 3*) results in more stable learning, when compared to a standard (linear) idealized model: $\tau = \tau_{max} u$. This experiment is repeated across different tasks, robot morphologies, learning/control algorithms, and simulated environments. The muscle morphology has better data efficiency and is more robust to hyper-parameters changes.


[1] Siebert, T., and C. Rode. "Computational modeling of muscle biomechanics." Computational modelling of biomechanics and biotribology in the musculoskeletal system. Woodhead Publishing, 2014. 173-204.

**Issues:**

In general, i would ask the authors to respond to my points above. Some specific issues are outlined below:

* I would encourage the authors to implement stronger baselines (e.g. impedance controller [2]) as part of their study. In other words, investigate if muscles provide any benefit compared to the current state of the art in robotics.
* The authors should discuss the real world applicability of their findings. What should roboticist take away from this study?

**Quality Of The Limitations Section:**

Limitations are addressed clearly

**Reviewer Expertise:**

3: The reviewer is fairly confident that the evaluation is correct

**Robotics Focus:**

Relevant but unlikely to deploy to hardware in near future

**Strengths And Weaknesses:**

**Strengths**
* This paper demonstrates how muscle based actuation models can help learn better controllers in a variety of different scenarios. This is a non-trivial insight that could guide further hardware development.

**Weaknesses**
* It’s unclear if the proposed benefits are actually unique to muscle morphology. The key issue is the baseline used in their study (ideal actuator) is very naive. Most robots already use impedance controllers [2], position controllers [3], and even learned controllers [4] that may provide similar benefits (e.g. “smooth control” constraints) without requiring custom hardware.
* The study doesn’t verify that key muscle properties (e.g. non-linearity) are actually important for better learning. For example, why couldn’t a well tuned low pass filter achieve the same level of smoothness? Thus, some of the claims in the paper (e.g. that non-linearity is a strength) are not properly backed up.
* Finally, this paper is purely demonstrated in simulation, and it’s unclear if its conclusions will generalize to the wild. Indeed, it’s likely that real muscles have properties that are difficult to simulate and/or deviate from the paper’s activation model in significant ways. This limits the study’s potential impact on practical robot learning applications.

[2] Hogan, Neville. "Impedance control: An approach to manipulation: Part II—Implementation." (1985): 8-16.

[3] Rocco, Paolo. "Stability of PID control for industrial robot arms." IEEE transactions on robotics and automation 12.4 (1996): 606-614.

[4] Martín-Martín, Roberto, et al. "Variable impedance control in end-effector space: An action space for reinforcement learning in contact-rich tasks." 2019 IEEE/RSJ International Conference on Intelligent Robots and Systems (IROS). IEEE, 2019.

**Summary Of Recommendation:**

This paper investigates a fascinating topic and provides interesting insight on where muscle morphology may outperform idealized torque actuators. However, the practical takeaways are muddled because the baseline is very naive, and the simulated results may not be indicative of real world performance/behavior. As a result, I don’t think this paper has quite reached the acceptance threshold, but I will keep my mind open throughout the rebuttal period.

---

> ### Author Response · Authors · 2022-08-22
> **PD Controller Results, low-pass filter experiments, nonlinearity experiment, applicability to real robotics discussion**
>
> We thank the reviewer for the comments and for the suggested improvements. We have addressed your comments and questions below:
> > Most robots already use impedance controllers [2], position controllers [3], and even learned controllers [4] that may provide similar benefits
>
> We thank you for these detailed suggestions. Due to time-constraints, we were able to implement only one of the suggested controllers as an additional baseline so far:
>  We used a position controller (PD) on top of the torque actuator morphology and repeated the learning tasks for point-reaching using RL and optimal control. For RL, we found that, while learning performance was adequate for singular runs, the mean performance was also not much better than for torque control (**Fig. 18, 19**).
>
> For OC (**Fig. 24**), we found that data-efficiency is slightly improved using an additional PD layer for the torque controller, however, the muscle actuator still achieves a better data-efficiency and is able to find a good-enough optimum with fewer iterations. For more details, we would like to refer to the main response to all reviewers.
> > The study doesn’t verify that key muscle properties (e.g. non-linearity) are actually important for better learning.
>
> To show-case that different key muscle properties and their non-linearity are important for better learning, we performed some additional experiments: The four major properties that differ between the torque actuator morphology and the muscle actuator morphology are the nonlinear activation dynamics, the nonlinear force-length, the nonlinear force-velocity relation and the nonlinear lever arms.
> We switched each of these properties off separately to test which nonlinear muscle property influences the beneficial behavior shown in the main paper the most. The results can be seen in **Fig. 23** in the Appendix.
>
> As shown in this figure, switching off the nonlinear force-velocity relation (no Fv) has the strongest impact and leads to results that are even worse than the torque actuator optimization. Additionally, the nonlinear activation dynamics has some influence on the performance of the data-efficiency results. With these results, we would like to give a first indication that indeed the non-linearity of different muscle properties are beneficial for the data-efficiency in learning anthropomorphic tasks.
>
> > For example, why couldn’t a well tuned low pass filter achieve the same level of smoothness?
>
> Additional to implementing a PD controller on top of the torque actuator morphology (see general response to all reviewers), we also implemented a low-pass filtered torque actuator. We tuned this low-pass filter for two different settings: One with a slow and one with a fast-filter response (the latter one uses similar parameters as the activation dynamics in the MuJoCo muscle model). Using these low-pass filters, we re-trained the point-reaching tasks and compared the learning performance to the original torque and muscle actuator morphology (shown in **Fig. 18 and 19 in the Appendix**). As shown in these figures, the muscle actuator morphology still outperforms all other considered actuator designs. This also applies to the robustness experiments, where an unknown chaotic weight was attached to the hand which was not present during learning (shown in **Fig. 20 and 21 in the Appendix**). Here again, the muscle actuator proves to be more robust for all considered masses compared to the other actuator designs.
>
> Regarding applicability to real-world robotics:
> We agree with the reviewer that in our original submission this point was not made very clear. Therefore, we included a new discussion paragraph to the main paper. Please, also find a more detailed answer in the general response to all reviewers.
>
> In summary:
> * We implemented stronger baselines, including PD controllers for the torque actuator morphology (**Appendix: RL Fig. 18, 19 second row, OC Fig. 24**) and low-pass filters (**Appendix: Fig. 18, 19 third and fourth row**)
> * We performed a variation study based on the nonlinearities in the muscle model (**App. Fig. 23**)
> * We added a discussion paragraph to the main paper (**section 5**) regarding real-world robotics.

---

### Official Review · Reviewer_XEHZ · 2022-07-30

**Originality:** Very Good
**Technical Quality:** Very Good
**Clarity Of Presentation:** Excellent
**Impact:** 4

**Recommendation:**

Strong Accept: I recommend accepting the paper and will argue for my recommendation even if other reviewers hold a different opinion.

**Summary:**

This paper performs a variety of experiments to investigate the benefits of replacing standard torque actuation with bioinspired muscle dynamics for learning-based control of multi-link rigid body manipulator-style system (i.e., humanoid systems). Across a wide variety of tasks and experiments, it is found that muscle morphologies can increase the learning rate and data efficiency, while also providing higher robustness to disturbances not seen during controller training. The paper also clearly notes that muscle morphologies are limited in that they are much more difficult to implement in both software and hardware, and that existing/classical torque control approaches can likely be tuned to attain similar performance. Overall, however, the results of this paper are interesting and compelling.

**Issues:**

Figure 1: The current illustration is odd because the control input branches off, then is recombined into the final torque. Perhaps it makes more sense to split this into two separate, parallel flowcharts, with the input $u$ and output $\tau$ the same for both?

14: The first sentence would really benefit from an example of a "complex anthropomorphic motion."

20: This paragraph could benefit from a reference to Fig. 1, to more fully explain the difference between the torque and muscle morphologies.

Fig 3: It is a bit difficult to tell the difference between the similarly-colored curves in the top row (but I think this will be hard to modify to make it clearer). It would also help to explain what $c$ is in the figure caption, since that parameter does not clearly appear in any equations, only in a single sentence of the text.

162: The phrase "vary much stronger" is a bit weird here. I guess this should be "vary much more," to avoid confusion with "stronger" referring to actuator effort.

209: "dependant" is spelled incorrectly

**Quality Of The Limitations Section:**

Limitations are addressed clearly

**Reviewer Expertise:**

4: The reviewer is confident but not absolutely certain that the evaluation is correct

**Robotics Focus:**

Relevant but unlikely to deploy to hardware in near future

**Strengths And Weaknesses:**

STRENGTHS

First, this paper is scoped well, and the writing quality is very good. The main text provides all details needed to clearly understand the story and key ideas. The supplementary content and videos provide useful additional details and visualizations.

Second, the experiments appear well thought out and thorough, and cover many representative anthropomorphic tasks. Due to this variety of experimental design, the numerical results are quite convincing.

Third, the paper is very up-front and clear about limitations and possible alternative interpretations of the numerical study. This makes the overall message more compelling, as it invites readers to further investigate the utility of muscle morphologies for themselves.

WEAKNESSES

The abstract should highlight the exciting takeaways of the extensive numerical experiments. Currently, it only states that the paper "investigates" the differences between torque and muscle actuator morphologies.

It would be nice to see how the torque and muscle morphologies perform across a range of mass changes, instead of just a single 1.5 or 1.0 kg change. I am inclined to believe that the muscle morphology really does have an advantage here, but I think the experiment can be made stronger.

The paper has no hardware experiment. This is a very minor weakness, as creating a muscle actuator morphology on hardware would itself be a very good CoRL paper.

**Summary Of Recommendation:**

I think this paper is very well scoped, and provides compelling evidence in support of a useful hypothesis for task-dependent robot actuator and control design. The experiments are extensive and thorough, and show a general trend of utility for the muscle actuators. Critically, muscle actuators are not presented as a silver bullet; instead, the limitations are discussed very clearly and give the paper good tutorial value for the robotics community.

---

> ### Author Response · Authors · 2022-08-22
> **Adapted abstract and figure 1, added perturbation experiments with more masses, PD controller, low-pass filters**
>
> We thank the reviewer for the overwhelmingly positive response! We are glad that the potential impact of our work became clear in the manuscript.
>
> > The abstract should highlight the exciting takeaways
>
> We have adapted the abstract to read less general and fit the results better.
>
> > how the torque and muscle morphologies perform across a range of mass changes, instead of just a single 1.5 or 1.0 kg change.
>
> We have added perturbation experiments with different masses for reaching experiments. For the RL results, we also provide results for the newly created PD and low-pass filter actuators. (**App. Fig. 18**). Additionally, we recorded new learning curves for larger maximum torque values, to see if a stronger underlying motor could help against perturbations. (**App. Fig. 19**)
>
> We observe that the muscle properties lead to better stabilization in ALL scenarios, even though it slightly undershoots the targets for very heavy objects. (**App. Fig. 20, 21**)
> While a stronger torque actuator improves the robustness, it also leads to slower learning, similar to our hopping force variations in **App. Fig. 13**.
> The muscle actuator achieves fast low-variance learning while its properties help counteract perturbations.
>
> Additional robustness tests were also performed for the MPC scheme comparing different weights for the torque and muscle morphologies shown in **App. Fig. 25**. We see that both actuators are able to counteract unknown perturbation weights with $1$ kg. For larger weights, the perturbations result in overshoots in the elbow joint angle which can be corrected in the muscle-actuated case, whereas the torque actuator struggles to counteract these perturbations. Summed up, the muscle morphology is more robust towards perturbations for a wide range of different unknown weights.
>
> > The current illustration is odd because the control input branches off
>
> We agree that the illustration was misleading and have adapted it.
>
> > The first sentence would really benefit from an example of a "complex anthropomorphic motion."
>
> We have added an example.
>
> > The phrase "vary much stronger" is a bit weird here
> "dependant" is spelled incorrectly
>
> We have implemented the suggestions and thank the reviewer again for such a detailed reading of our text.
>
> >  It is a bit difficult to tell the difference between the similarly-colored curves (but I think this will be hard to modify to make it clearer) It would also help to explain what  $c$ is in the figure caption
>
> It is indeed difficult to adapt these colors while retaining a common pattern. We have adapted the caption following your suggestion.
>
> In summary:
> * We have provided new results for a wide range of weight perturbations in the point-reaching tasks (**Appendix: RL Fig. 20, 21; OC Fig. 24**)
> * We fixed all minor spelling, formulation and graphics suggestions (**throughout main paper**)

---

### Official Review · Reviewer_SnhT · 2022-07-31

**Originality:** Good
**Technical Quality:** Very Good
**Clarity Of Presentation:** Good
**Impact:** 3

**Recommendation:**

Weak Accept: I recommend accepting the paper, but will not argue for my recommendation if the majority of other reviewers have a different opinion.

**Summary:**

This paper evaluates the impact of using a muscle actuator model as an action space for learning-based control. Data efficiency, hyperparameter sensitivity, and robustness of the learned behavior are evaluated on a set of simulated robotics tasks. The results indicate some empirical benefits of muscle actuation over direct torque control.

**Issues:**

- A related work is notably omitted: “Learning Locomotion Skills Using DeepRL: Does the Choice of Action Space Matter?” (Peng et al. 2017 / https://arxiv.org/pdf/1611.01055.pdf). Here, the authors compared Torque, MTU, position, and velocity action spaces regarding data efficiency and robustness on several RL locomotion tasks. This paper may have substantial differences from the results presented here but the distinction is currently unclear to me. Please add information to elucidate the differences, so that I can better assess the impact and originality of this work.
- It's much more common to use proportional-derivative control than direct torque control in learning motor skills for robots. This is because position targets are a more efficient, robust, action space than torque control. Why not make this comparison in your experiments? It's likely to be more interesting to roboticists than a comparison to torque control.
- In future work, I would also be curious about how model-based RL performs in the learning with muscles setting, given that it potentially needs to learn a more complex forward dynamics model. This could make for some interesting analysis. But this is just a suggestion and by no means needs to be addressed here.
- Another good reference to consider for the impact of environment design on algorithm performance is Learning to Locomote: Understanding How Environment Design Matters for Deep Reinforcement Learning (Reda et al 2020 /  https://www.cs.ubc.ca/~van/papers/2020-MIG-envdesign/2020-MIG-envdesign.pdf)

**Quality Of The Limitations Section:**

Limitations are addressed clearly

**Reviewer Expertise:**

4: The reviewer is confident but not absolutely certain that the evaluation is correct

**Robotics Focus:**

Relevant but unlikely to deploy to hardware in near future

**Strengths And Weaknesses:**

Strengths: The analysis is thorough and the premise of bio-inspired action space design is interesting. I learned about some empirical trends for optimization with MTUs vs torque control.

Weaknesses: The paper should address its relationship to some related prior work. Also, the takeaways for robot learning are ambiguous without comparing to the commonly used PD control action space.

**Summary Of Recommendation:**

The paper is technically sound, and the question of action space design for learning-based control is interesting. However, uncited previous work seems to have provided a similar set of results. The authors should clarify how their work differs. Also, to make a valuable paper for robot learning practitioners, the authors should add the PD control action space as a baseline.

---

> ### Author Response · Authors · 2022-08-22
> **Additional discussion regarding Peng et al., results for PD controller**
>
> We thank the reviewer for their excellent paper suggestions! We really enjoyed reading the related works.
>
> > A related work is notably omitted
>
> Peng et al. provided an interesting comparison of actuator performance in walking tasks and used a muscle model as one of the baselines. They found PD-control to often outperform pure torque-control, while their muscle actuators did neither lead to faster learning, nor to better robustness. While this seems to contradict our results, we believe there is a very probable reason for this: Peng et al. trained the policies to imitate reference trajectories as closely as possible, while we learn behaviors from scratch to solve a task.
>
> Fundamentally, a naive torque controller can ALWAYS provide trajectory tracking at least as well as a muscle actuator and often better. The output of muscle actuators is always dependent on the current fiber lengths and velocities, making it trajectory-dependent. As such, it is clear that it might not be able to track every imaginable trajectory.
> To further our point, please take a look at **Fig.8** (**main paper**): The learned muscle and torque trajectories are vastly different. While muscles might not be able to imitate the torque trajectory, they help with learning in more general and unstructured tasks.
>
> Additionally, the reward terms and state input in [1] were strongly focused on replicating given positions over time which is more closely aligned with position control, explaining the dominance of the PD-controller. Your other paper suggestion [2], also argues that PD-control might be much better for reference tracking tasks than learning from scratch. Thank you for the advice!
>
> We also believe that the worse robustness of muscle actuators in [1] was due to the optimization procedure: The parameters of the muscles were tuned to replicate the reference trajectory as closely as possible, which might constitute “over-fitting” to the trajectory, leading to bad generalization. Our more general muscle models seem to perform much better in this regard.
>
> Finally, we also performed experiments with MPC, not just RL, used two muscle models of different complexity, provided reaching, hitting and hopping tasks, gave extensive hyperparameter evaluations and gave results for a complex 3D humanoid (Fullbody model: allmin). All of these aspects go beyond Peng et al. and demonstrate that muscles can provide benefits for learning and robustness when learning behaviors from scratch.
> We added a discussion to the main paper, section 1.
>
> > more common to use proportional-derivative control than direct torque control
>
> We have added several experiments with PD-control on the reaching task for both the reinforcement learning as well as the optimal control (OC) case as additional baseline. The idea behind this baseline was to use PD control on top of the torque actuator morphology. For RL, we found that, while learning performance was adequate for singular runs, the mean performance was also not much better than for torque control (**Fig. 18 Appendix**). For OC (**Fig. 24 Appendix**), we found that data-efficiency is slightly improved using an additional PD layer for the torque controller, however, the muscle actuator still achieves a better data-efficiency and is able to find a good-enough optimum with fewer iterations.
>
> In summary:
> * We have provided a discussion of the differences of [1] to our work here and to the main paper. (Section 1)
> * We have added results with a PD controller (**Appendix: RL Fig. 18, 19, OC Fig. 24**)

---

> > ### Comment · Reviewer_SnhT · 2022-08-26
> > **Response from Reviewer SnhT**
> >
> > Thanks for the response. Your added experiments and revisions to the paper address my concerns. I now believe the information in this paper is accurate, appropriately contextualized and would benefit the community. I will raise my rating to a weak accept (although I don't have an option to edit the original review until the review period ends).

---

> > > ### Author Response · Authors · 2022-08-27
> > > **Thank you for your positive response!**
> > >
> > > We thank you for reviewing our paper and taking the time to read the updates! We appreciate the feedback and think that the suggestions definitely improved our submission.

---

### Official Review · Reviewer_vcXi · 2022-07-31

**Originality:** Fair
**Technical Quality:** Good
**Clarity Of Presentation:** Good
**Impact:** 2

**Recommendation:**

Weak Accept: I recommend accepting the paper, but will not argue for my recommendation if the majority of other reviewers have a different opinion.

**Summary:**

This is a benchmarking paper evaluating muscle-like control against vanilla torque control in simple, anthropomorphic tasks. There are two kinds of muscle models deployed - the demoa model and a much simpler model written in Mujoco. There are three types of control algorithms being tested, optimal control (trajectory optimization) with CMA-ES, MPC, and DRL with MPO. The paper shows that vanilla torque control often leads to a very large variance across random seeds, therefore hurting its mean performance, and robustness against perturbation; while muscle-like control often manifests a much smaller variance and better robustness.

**Issues:**

- Comparison with setting spring & damping in each simulated joint for torque control - could the authors at least discuss why they think this will be insufficient to bring the performance close to muscle-like controls?

- Comparison with using torque rate control: it does not seem very fair to compare muscle control to torque in the sense that muscle control do not change activation level directly. A more fair action space to compare against might be “delta torque”, or “torque rate“, where delta torque at each step is added to the previous executed torque.

- Ablation of using smaller simulation time steps - see weaknesses.

- Metrics are simply losses which include actions - see weaknesses.

- Why is number of random seeds for each task sometimes 5, sometimes 8?

**Quality Of The Limitations Section:**

Limitations are addressed clearly

**Reviewer Expertise:**

4: The reviewer is confident but not absolutely certain that the evaluation is correct

**Robotics Focus:**

Relevant but unlikely to deploy to hardware in near future

**Strengths And Weaknesses:**

Strengths:

- The authors seem very honest by reporting a lot of details in their experiments, and providing code (though I don’t have the chance to take a look at the code)

- Though number of tasks evaluated is not great, and the tasks are somewhat simplistic, the paper covers two models and three control modalities which I believe is a plus.

Weaknesses:

- Looking at the result figures, torque control is not always bad, just has a really large *variance* across random seeds. But the paper does not give much insight on why this is the case - for example, state-of-the-art DRL algorithms seem to be quite random-seed-robust on simple tasks such as hopping or reaching even with torque control. Why is such *robustness across seeds* absent in the experiments here?

- Related to the point above, I’m not sure if the difference in performance actually mainly comes from *simulation & numerical stability*. The authors mentioned that they have to make the time-step quite large (0.01s) to produce noticeable differences between control schemes. So what happened when time-step is in the usual range of stable physics simulations, e.g. 250~1000Hz? (See also Issues, for additional ablations)

- Is using final loss as the main performance metric fair? The losses of several tasks include action squared, which means totally different quantaties for muscle and torque control. I seem to believe the metric should always be action-space-agnostic.


**Summary Of Recommendation:**

(Edit 08/28: most of my questions and doubts are decently addressed by the rebuttal and revision. Bumping my score to Weak Accept.)

This paper is a good benchmarking effort. While I believe it should be eventually published and shared somewhere, I’m not sure if CoRL is the best fit. Also given the weaknesses, the paper might also not reaching the bar of CoRL.

My main concern, besides doubting whether the CoRL community would care enough about this topic, is what these experiments really tell us. As aforementioned, an alternative conclusion that one could draw from these experiments can be - we need to make simulation more numerically stable, by for example adding springs and dampers to each torque-controlled joint, which is quite standard in most simulation environments, or by not making simulation frequency too low.

---

> ### Author Response · Authors · 2022-08-22
> **experiments with different simulation time steps and explanation, new learning and perturbation experiments with PD and low-pass filter and agnostic action-space**
>
> We thank the reviewer for the detailed comments regarding our methodology.
>
> > simulation & numerical stability
>
> We apologize for our unclear description, numerical stability was never an issue in our experiments.
> There are two different timesteps: the physics simulation and the control time step. We used simulation time steps of 0.005s in the fixed step solver of MuJoCo while the demoa time step is set by an adaptive step size integrator (Shampine/Gordon) with a maximum time-step of 0.001s. To demonstrate that numerical stability is not an issue even in the 200Hz MuJoCo environment, we ran the ArmMuJoCo environment with smaller step-sizes while keeping the control time step equal (see **Fig. 22 App.**).
>
> The second larger time scale is the control update time step. In our MPC and RL scheme algorithm, this is set to 0.01s. This means that every 0.01s the learning algorithm selects a new optimal control signal u which is then kept constant for the next 0.01s (while the physics are solved with the small adaptive time steps).
>
> We added the following sentence to the methods section: Physics differential equations were solved with variable time step (max time step 0.001s) in Demoa and fixed time step (0.005s) in MuJoCo.
>
> We hope we could convince you of the numerical stability of the environments.
>
> > Setting spring & damping in each joint for torque control
>
> We took up this idea and implemented a PD controller (mimicking elastic and damping characteristics) on the reaching tasks as another comparison environment.(**Appendix: Fig. 18 for RL, Fig 24 for OC**). We refer to the main comment for details. We still observe the same benefits for muscles.
>
> > Comparison with using torque rate control
>
> We appreciate the idea of torque-rate control as another option. However, we want to clarify that the muscle did not specify actions as desired changes in activity.
>
> The muscle activation dynamics (Eq. (2)) simply implement a low-pass filter on the control signal u (inspired by muscular biochemical processes). We show such low-pass filters in **Fig. 17 Appendix**.
>
> To keep the comparison between torque actuator and muscle fair, we ran new experiments where we also applied a low-pass filter to the torque actuator (see **Fig. 18, 19 Appendix** and the main comment).
>
> Additionally, we conducted perturbation experiments with all new controllers (PD, low-pass), see **Fig. 20, 21 App.**), and found none to match the muscle stabilization.
>
> > I seem to believe the metric should always be action-space-agnostic
>
> The smooth point-reaching task, the ball-hitting task, the squatting and the high-jumping task were specifically designed to be action-space agnostic. Originally, we included action terms in the fast/precise point-reaching and the hopping task as regularization terms. However, due to your comment, we carefully checked again and found the contribution of action costs to be insignificant when plotting.
> Nevertheless, we reported all new experiments with distance returns without including the action cost in the **App. (Fig. 18-22**).
>
> > Number of random seeds is sometimes 5 and sometimes 8
>
> This was a pragmatic choice: for the MPC simulations, we did not have access to a compute cluster and therefore limited the number of random seeds.
>
> > torque control is not always bad, just has a really large variance across random seeds.
>
> This is an interesting observation!  We now additionally show learning curves for individual runs in **Fig. 18, 19 App.**. We observe that indeed, there is a much larger variance across seeds for all torque actuators! However, most muscle actuator runs still perform better than the best torque actuator run.
>
> We also want to point out that it is not uncommon to observe such variance across seeds in the literature. See [1], which identifies a one-legged hopper as exhibiting one of the largest variances across seeds in all deepmind control suite tasks.
> We also want to reference the MPO paper [2] (p.11 Fig.4): we see that many RL algorithms exhibit very large seed variance across different tasks.
>
> Finally, while we encounter large variance across seeds in our investigation, which might be remedied with simulation settings for some tasks, this might not be the case for others. But If  such a variance can be observed, muscle-like properties seem to reduce the effect.
>
> In summary:
> * We performed learning experiments with different simulation time steps (**App. Fig. 22**)
> * We added learning (**App. Fig 18, 19**) and perturbation experiments (**App. Fig. 20, 21**) with a PD controller and low-pass filters for the torque actuator. (**App. Fig. 24 for OC**)
> * We give these results with the distance return, without action costs.
> * We have hopefully addressed all remaining points in the text.
>
> Papers:
> [1] Bjorck, J., et al. "Is High Variance Unavoidable in RL? A Case Study in Continuous Control." arXiv preprint 2021.
> [2] Abdolmaleki, A., et al. "Maximum a posteriori policy optimisation." arXiv preprint 2018.

---

> > ### Comment · Reviewer_vcXi · 2022-08-25
> > **A lot of added experiments & ablations**
> >
> > I much appreciate the authors' effort to improve the paper within such tight time frame. While a lengthy paper don't usually correlates with high quality, number of experiments & ablations are certainly crucial for such a benchmarking paper.
> >
> > I believe the new experiments add value to the submission, and readers could learn more from this revised version. I'm **leaning positive** now for this submission, if the authors could kindly answer the following questions and consider including the answers in the final version:
> >
> > - The rebuttal mentions "The muscle activation dynamics (Eq. (2)) simply implement a low-pass filter on the control signal u". So why low-pass filters don't help torque control (Fig. 18) at all but activation dynamics is important for the performance of muscle control (Fig. 23)?
> >
> > - As mentioned in rebuttal "... contribution being insignificant", could the authors promise to re-plot all figures in main text without contributions of the action loss term?
> >
> > - Could the authors promise to open-source code for all experiments if the paper is accepted?
> >
> > - Could the authors promise to re-organize the paper if it is accepted? Some of the important experiments (e.g. PD control, pertubation) could be merged into main text and repeated for all environments, while some interesting ablations (e.g. different components of muscle dynamics, increased torque limit) should be referenced in main text in for example the format of an inventory list, to draw attention of interested readers. Some other experiments not mentioned by any reviewer could potentially be moved to Appendix, on the other hand.

---

> > > ### Author Response · Authors · 2022-08-26
> > > **Thank you for your positive response!**
> > >
> > > We thank you very much for taking the additional time to read the rebuttal and are excited that we could convince you of the value of our work. Regarding your remaining comments:
> > >
> > > > So why low-pass filters don't help torque control (Fig. 18) at all but activation dynamics is important for the performance of muscle control (Fig. 23)?
> > >
> > > Thank you for noting this interesting observation. Indeed, it is difficult to compare these two figures (Fig. 18 and Fig 23) directly: In Fig 18, we included a simple low-pass filter as used in the MuJoCo muscle model (Eq. 9 in the Appendix). Including this simple low-pass filter did not lead to significant benefits for the torque actuator. In contrast, in Fig. 23 we have switched off different muscle properties including the nonlinear activation dynamics while still keeping all other parts of the remaining muscle dynamics. We hypothesize that there is still an interplay with the remaining muscle dynamics that ultimately helps performance.
> > > Furthermore, we would like to point out that in Fig. 23 the more complex muscle model (demoa), and therefore, also more complex activation dynamics, were used (Eq. 18 in Appendix). This Hatze activation dynamics is much more complex than commonly used low-pass filters (as were  implemented in Fig. 18) and for example, depends on the **muscle fiber lengths**. As the torque actuator does not have such a muscle-length dependency, it is impossible to directly include this Hatze activation dynamics as a low-pass filter.
> > >
> > > We admit that it is not always clear which plot belongs to which simulator and muscle model, but we will put additional (RL) (OC/MPC) labels on each plot to ensure the distinction. Also, we will add more discussion about these differences to clear up any confusions in the current text.
> > >
> > > > As mentioned in rebuttal "... contribution being insignificant", could the authors promise to re-plot all figures in main text without contributions of the action loss term?
> > >
> > > We agree that a performance comparison should not include the action term in order to be scientifically rigorous. We will replot the experiments in the main paper upon acceptance.
> > > > Could the authors promise to open-source code for all experiments if the paper is accepted?
> > >
> > > This was on our agenda from the beginning and should have been mentioned. We promise to open-source the code for all experiments. Gladly, both simulators are freely available already.
> > > > Could the authors promise to re-organize the paper if it is accepted?
> > >
> > > We agree that some of the results that came up in the rebuttal deserve to be in the main paper, as they make the results much more compelling. We promise to re-evaluate which experiments belong to the main paper and we will do our best to extend the new analysis to the existing environments.
> > > We will try to summarize results that were not mentioned by reviewers into tabular form free up space for other, more important experiments. Of course, we also have to add references to the results that were added in the appendix, instead of just attaching them.
> > >
> > > We thank the reviewer again for giving the updates a second read! We appreciate the feedback and think this will definitely improve our submission.

---

### Author Response · Authors · 2022-08-22
**Revised Submission and Supplementary**

We have uploaded a revision of the main paper and the supplementary in this document as a single PDF. Changes are marked in red.

---

### Author Response · Authors · 2022-08-25
**Thank you and we are happy to discuss further!**

We would like to thank all reviewers again for their valuable feedback and very much appreciate their assessment of our work as “a good benchmarking effort” (Reviewer vcXi), with “thorough analysis” (Reviewer SnhT) based on “interesting and compelling results” (Reviewer XEHZ). All reviewers were confident that this paper investigates a fascinating topic and pointed out that it offers “a non-trivial insight that could guide further hardware development” (Reviewer KLzq).

As the rebuttal is coming to an end, we would like to point out again that we are happy to discuss any remaining comments or suggestions. If everything was resolved to your satisfaction, we would be very thankful if the review scores would be adjusted accordingly.

---

### Meta-Review · Area_Chair_DdNp · 2022-08-10

**Recommendation:** Accept (Poster)
**Confidence:** 4

**Metareview:**

This paper compares the learning performance of anthropomorphic systems actuated by physiological muscles as well as typical robotic controllers such as PD position control and direct torque input. The simulation results demonstrate that the nonlinear muscle model is superior in terms of data efficiency, sensitivity to hyperparameters, and robustness of the resulting policy in many test cases.

The presented idea is inspiring and potentially has deep implication to machine learning and robot design, while practical application to robot hardware may be questionable at least in the short term. Nevertheless, the paper will likely stimulate active discussions on how the choice of robot actuators may affect robot learning performance.

**Best Paper Nomination:**

No

---

> ### Author Response · Authors · 2022-08-22
> **PD Controller Results, low-pass filter experiments, nonlinearity experiment, and applicability to real robotics discussion**
>
> We thank all the reviewers and the AC for their insightful comments so far! The most common concerns relate to (A) the applicability of our findings to real-world robotics and (B) the comparison to common baselines such as a PD controller. We first give a summary of our revisions before we address these two points here in detail. The revised document is uploaded, for your convenience, we’ve attached the appendix.
>
>
> Summary:
> * We have given new results with PD-controllers and low-pass filters for RL (**Fig. 17-19 Appendix**) and with a PD-controller for OC (**Fig. 24**)
> * We performed a simulation time step ablation for MuJoCo (**Fig. 22**)
> * We performed perturbation experiments with more masses (**Fig. 20, 21 RL, 25 OC**)
> * Muscle property ablations hint at the importance of the nonlinearities for the reported benefits (**Fig. 23**)
> * Added a paragraph about robotic applicability in the discussion (**Sec. 5**)
> * Added a discussion about [7]. (**Sec. 1**)
> * Slightly changed the abstract, adapted **Fig. 1**
>
>
> (A) Applicability to real robotics
>
> We believe the simplified MuJoCo muscle model can already be applied to real robotic systems. Just like a PD-controller, its force dynamics can be simulated and integrated in low-level control layers applied to a real system. Furthermore, this has already been done for some muscle properties [1,2,3,4]. The only requirements are a large control frequency and fast computation of the model. As our MuJoCo muscle model is quite simplistic, it should be possible to simulate its output for a real robot.
>
> We agree, however, that a biomechanically accurate hardware muscle is not feasible at the moment. Nevertheless, many researchers work on simplified soft actuators that exhibit some muscular properties and might one day revolutionize applications such as wearable rehabilitation devices [5] and or improve the design and control of humanoid robots and assistive exoskeletons in terms of robustness, safety and energy-efficiency [6]. While soft actuated systems are hard to control from a classical control theory point of view, our results and other works [7] suggest that RL may even benefit from their properties.
> We believe that our results with demoa, with its complex muscle models, might give roboticists an idea of what could be gained from soft actuators in terms of learning and optimization.
> For further details, we would like to point to our new discussion paragraph in the main paper.
>
> (B) PD-controller
>
> For the RL setup, we used a PD implementation identical to [7]. The gains were tuned to produce stable positions for the whole work space of the ArmMuJoCo robot, as shown in **Fig. 8 Appendix**. We also added two torque-variants which use a low-pass filter, identical to the one used in the MuJoCo muscle model, with two different filtering settings.
> We retrained point-reaching (**Fig. 18 Appendix**) and show average learning curves and performance of all individual runs. We did the same with larger maximum torque limits for the torque actuators and show the result in **Fig. 19 Appendix**.
> The muscle actuator outperforms all other controller in the reaching task.
>
> With these actuators and learned policies, we also performed new perturbation experiments with different masses. Again, the muscle actuator outperforms all baselines.
>
> For the OC setup, we also included a PD implementationl on top of the torque actuator morphology instead of only considering the torque action space. The data-efficiency results for this experiment are shown in **Fig. 24 Appendix**. Here, we show the averaged learning curves of all individual runs. Similar to the RL case, the muscle actuator still outperforms all baselines  even though the PD controller slightly improves the data-efficiency of the torque-actuator.
>
> [1] Garcia-Cordova, F., et al. "Emulation of the animal muscular actuation system in an experimental platform." 2001 IEEE International Conference on Systems, Man and Cybernetics. e-Systems and e-Man for Cybernetics in Cyberspace
> [2] Seyfarth, A., et al. "Simulating muscle-reflex dynamics in a simple hopping robot." Autonome mobile systeme 2007.
> [3] Knüsel, J., et al. (2020). Reproducing five motor behaviors in a salamander robot with virtual muscles and a distributed CPG controller regulated by drive signals and proprioceptive feedback. Frontiers in neurorobotics
> [4] Rai, A., et al.(2018, May). Bayesian optimization using domain knowledge on the ATRIAS biped. In 2018 IEEE International Conference on Robotics and Automation (ICRA)
> [5] Zhu, M., et al. "Soft, Wearable Robotics and Haptics: Technologies, Trends, and Emerging Applications." Proceedings of the IEEE 110.2 (2022)
> [6] Vanderborght, B., et al. "Variable impedance actuators: A review." Robotics and autonomous systems 61.12 (2013)
> [7] Peng, X., et al. "Learning locomotion skills using deeprl: Does the choice of action space matter?." Proceedings of the ACM SIGGRAPH/Eurographics Symposium on Computer Animation. 2017.*